# Tackling Inflammatory Bowel Diseases: Targeting Proinflammatory Cytokines and Lymphocyte Homing

**DOI:** 10.3390/ph15091080

**Published:** 2022-08-30

**Authors:** Yijie Song, Man Yuan, Yu Xu, Hongxi Xu

**Affiliations:** 1School of Pharmacy, Shanghai University of Traditional Chinese Medicine, Shanghai 201203, China; 2Shuguang Hospital, Shanghai University of Traditional Chinese Medicine, Shanghai 201203, China

**Keywords:** inflammatory bowel disease, cytokines, lymphocyte homing

## Abstract

Inflammatory bowel diseases (IBDs) are characterized by chronic inflammatory disorders that are a result of an abnormal immune response mediated by a cytokine storm and immune cell infiltration. Proinflammatory cytokine therapeutic agents, represented by TNF inhibitors, have developed rapidly over recent years and are promising options for treating IBD. Antagonizing interleukins, interferons, and Janus kinases have demonstrated their respective advantages in clinical trials and are candidates for anti-TNF therapeutic failure. Furthermore, the blockade of lymphocyte homing contributes to the excessive immune response in colitis and ameliorates inflammation and tissue damage. Factors such as integrins, selectins, and chemokines jointly coordinate the accumulation of immune cells in inflammatory regions. This review assembles the major targets and agents currently targeting proinflammatory cytokines and lymphatic trafficking to facilitate subsequent drug development.

## 1. Introduction

Inflammatory bowel diseases (IBDs) are a class of chronic inflammatory disorders including two main clinical entities: Crohn’s disease (CD) and ulcerative colitis (UC). The prevalence of IBD is increasing worldwide, imposing a huge socioeconomic burden on society and health care systems [1]. However, a thorough comprehension of IBD pathogenesis remains elusive, and the progression of IBD is widely believed to encompass an intertwining of environmental, genetic, microbial, and immunological factors [2]. Encouragingly, considerable advances in the development of drugs targeting the immune system to block IBD have boosted the efficacy of treatment for IBD in recent decades [3].

Overwhelming cytokine storms and immune cell infiltration impede inflammation regression in IBD and contribute to disease recurrence and tissue damage. As a chronic inflammatory disease, CD has been considered to be correlated with the immune response mediated by type 1 T helper cells (TH1) and TH17, while abnormal TH2 responses are involved in the development of UC, and the eventual imbalance of interactions with other T-cell groups (e.g., Treg and TH9) contributes to the complexity of IBD immunopathogenesis [2]. A variety of cytokines produced by immune cells are uniquely characteristic of patients with IBD, including tumor necrosis factor (TNF), interleukin (IL)-6, and IL-1β, which are critical drivers of inflammatory impairment (Figure 1) [4]. Moderation of such proinflammatory cytokines in IBD appears to be a sound treatment option, evidenced by the successful application of TNF inhibitors, and recognition of cytokines could facilitate IBD therapy. Furthermore, either the innate or adaptive immune response depends primarily on lymphocyte homing to specific tissues, which is mediated by the adhesion of immune cells to vascular endothelial cells [5]. Thus, modulating immune cell trafficking is an efficacious approach to remedy tissue lesions and mucosal inflammation in IBD.

Given the advanced research on proinflammatory cytokines and lymphatic trafficking associated with IBD in recent years, therapeutic agents for related targets are burgeoning. Therefore, a timely review of targeting cytokines and lymphatic trafficking is needed, and here, the latest relevant studies are collated to facilitate drug developers and clinicians to catch up on research developments and medication choices.

## 2. Targeting Proinflammatory Cytokines

Cytokines are pivotal influencers of gut homeostasis and perpetual inflammation [6]. The appearance of anti-TNF drugs has drastically altered the therapeutic algorithm for IBD, yet a significant proportion of patients receiving anti-TNF therapy suffer from nonresponse or secondary loss of response. Fortunately, new treatment alternatives have been proactively explored (Figure 2), among which some have entered clinical practice and some have shown good efficacy in preclinical studies [7].

### 2.1. Tumor Necrosis Factor Superfamily

The tumor necrosis factor superfamily (TNFSF) members are molecules with broad-spectrum activity and are well known to have pathological effects on IBD, the best known of which is TNF. The majority of TNFSF members are expressed by immune cells and modulate the homeostasis of T-cell-mediated immune responses [8]. Numerous recent studies have demonstrated that certain TNFSF members, including TNF-α, TL1A, FasL, and others, promote IBD pathogenesis through the enhancement of T-cell proinflammatory function and through the direct disruption of intestinal epithelial integrity [9].

#### 2.1.1. Anti-TNF-α Agents

Infliximab (IFX) is an immunoglobulin G1 monoclonal antibody (mAb) with a high affinity for TNF-α that was initially licensed in 1998 by the FDA for treating patients with refractory active IBD, pioneering the era of biologics for IBD [10,11]. IFX is potent in both attaining and sustaining clinical remission and is also recommended for patients with IBD tha are unresponsive to 5-ASAs or corticosteroid therapy. A multiple-center randomized controlled trial (RCT) of pediatric CD patients [12] showed that first-line IFX intervention outperformed routine treatment in terms of short-term clinical (59% vs. 34%) and endoscopic remission (59% vs. 17%). Sequentially, more anti-TNF mAbs, such as adalimumab, certolizumab pegol, and golimumab, have been developed and licensed, showing promising clinical efficacy [13]. For example, in a multicenter, prospective cohort study of CD patients, approximately two-thirds of patients receiving adalimumab had a good prognosis and long-term maintenance of remission, reducing the incidence of colectomy [14]. In addition to intravenous monotherapy, a new oral polyclonal anti-TNF antibody, AVX-470, prepared from the colostrum of TNF-immune cows, showed efficacy comparable to that of oral prednisolone and parenteral etanercept in murine experimental colitis models and minimized systemic exposure to anti-TNF agents [15]. The first RCT of AVX-470 for UC demonstrated a good safety profile and was beneficial for refractory UC, with clinical, endoscopic, and biomarker (serum CRP and IL-6) improvements [16]. Additionally, TNF-related biosimilars, such as CT-P13 and SB2, have been developed to mitigate the surge in therapeutic costs caused by biological treatments [17], increasing the accessibility of appropriate biologic treatments by patients. Currently, over 20 biosimilars of infliximab and adalimumab are under development and have been described in detail [18].

#### 2.1.2. Anti-TL1A Therapy

TNFSF member 15 gene (TNFSF15, also named TL1A) variants are correlated with the risk of IBD, which impacts the production of multiple cytokines to fuel mucosal inflammation [19]. TL1A is overexpressed in inflamed mucosa and is associated with CD fibrosis [20]. As TL1A is upstream of the proinflammatory process, anti-TL1A regimens may offer an attractive option for individuals unresponsive to TNF-α inhibitors and are a candidate for resolving currently nonreversible intestinal fibrosis. TL1A-overexpressing mice developed idiopathic ileitis with increased proximal colitis-inducing injury and fibrosis, whereas TL1A receptor-deficient mice exhibited resistance to a transmigration model of colitis and had reduced inflammation and intestinal fibrosis [21]. A human immunoglobulin G1 mAb against TL1A, PF-06480605, demonstrated improved endoscopic healing (38.2% at week 14, *p* < 0.001, and 95% CI = 23.82–53.68) and a favorable safety profile among individuals with refractory UC [22], probably in association with inflammatory T cells and fibrosis pathways [23]. C03V, a high affinity and selective human antibody to TL1A that neutralizes the biological activity of TL1A, remarkably attenuated the pathology of colitis triggered by trinitrobenzene sulfonic acid (TNBS) and alleviated fibrosis [24].

#### 2.1.3. Targeting the FasL/FAS System

Mutations in TNFRSF6 (also named FasL) and its receptor FAS cause autoimmune lymphoid tissue proliferation syndrome and are involved in lymphocyte variation in IBD [25]. A marked increase in FasL-containing cells was observed in active UC patients in comparison to either remission or control groups based on immunohistochemistry of the colonic mucosa [26]. Conversely, a study on an acute colitis rat TNBS model discovered that isolated dendritic cells from mesenteric lymph nodes of wild-type rats expressed more FasL than those from colitis rats, and exogenous infusion of DCs genetically engineered to overexpress FasL decreased T-cell IFN-γ production and enhanced T-cell apoptosis, effectively reducing colonic inflammation [27]. Osthole restored the downregulation of Fas and FasL expression in bone marrow mesenchymal stem cells (BMSCs) caused by immune impairment and restored the ability of BMSCs to induce T-cell apoptosis and immunosuppressive function over experimental colitis [28].

#### 2.1.4. Other TNFSF Members with Potential Efficacy

We are gaining insight into the value of other members of TNFSF in the IBD process as research progresses. TNFSF14, also named LIGHT, resides on T cells and participates in their activation, and its transgenic expression on murine T cells drives pathogenic inflammation in several organs, including the intestine [29]. Mice with Tnfsf14 deficiency were found to undergo more serious colitis and have lower survival rates than wild-type mice, suggesting that LIGHT may mediate innate immune activities and the resilience of gut inflammation by transmitting signals through lymphotoxin β receptors in the colon [30]. Furthermore, single-cell analysis of colonic mesenchymal cells in UC patients also revealed a subpopulation of activated mesenchymal populations with TNFSF14 expression, which is assumed to limit colonic epithelial proliferation and impede wound repair responses [31]. TNFSF10, also named TRAIL, mediates autoimmune inflammation and cellular homeostasis by transmitting apoptotic signals to induce apoptosis and was found to suppress autoimmune colitis by inhibiting colonic T-cell activation [32].

It is worth noting that not every patient responds to anti-TNF therapy and many suffer from the underlying opportunistic infections that come with treatment. High-dose administration regimens are recommended by experts to address the insufficient efficacy of anti-TNFs, and individualized dosing strategies are encouraged to be implemented with drug monitoring in clinical practice to avoid inadequate blockade of TNF in serum and tissues [33]. Natural products and herbs have also shown modulation of TNF-α activity in experimental models and clinical trials, which might be an effective intervention to bypass the side effects of specific anti-TNF therapy [34]. Natural polyphenols from plants, such as curcumin [35], mangiferin [36], and catechin [37], have demonstrated anti-inflammatory effects in TNBS/DSS-induced colitis models and reduced TNF-α expression. In addition, complementing anti-TNF therapy with nutritional biocompounds, such as antioxidant-enriched purple corn supplement [38], has exhibited beneficial effects on the induction and maintenance of IBD remission. While studies on natural products are currently mostly preclinical, their potential for the treatment and prevention of IBD deserves constant attention.

### 2.2. Interleukins

Multiple interleukins secreted by lymphocytes mediate their pro/anti-inflammatory effects, and inhibition of proinflammatory cytokines (IL-6, IL-1β, IL-12, and IL-23) is an effective strategy to ameliorate colitis. Presently, agents targeting interleukins have exhibited some advantages over anti-TNF agents.

#### 2.2.1. Anti-IL-12/IL-23 Agents

IL-12 and IL-23, produced mainly by inflammatory myeloid cells, are highly expressed in colitis patients and induce differentiation and responses in TH1 cells and IL-17-producing T helper cells (TH17), respectively [39]. IL-12 and IL-23, as heterodimeric cytokines, are formed by the homogeneous subunit p40 in conjunction with p35 and p19, respectively. The roles of IL-12 and IL-23 are temporally distinct in the progression of IBD, where IL-12 is engaged in the early stages of colitis by activating TH1 cells and macrophages, while IL-23 shapes the chronic exacerbation of the disease [40].

Antagonizing p40, the cosubunit for IL-12 and IL-23, is a potent blockade of disease development. Ustekinumab (UST), a mAb against the p40 subunit, has already been licensed for CD treatment. In two 8-week RCTs of intravenous induction therapy for intermediate-to-severe active CD patients (UNITI-1 and UNITI-2), patients undergoing either primary or secondary nonresponse to TNF antagonists achieved significantly higher remission rates in patients receiving intravenous UST than those receiving placebo (in UNITI-1, 34.3% vs. 21.5%; in UNITI-2, 51.7% vs. 28.7%). A double-blind, active comparative phase 3b RCT (SEAVUE) involving multiple countries showed that 124 (65%) moderate-to-severe CD patients (n = 191) without biologic therapy remained in clinical remission after UST treatment at week 52 [41]. Moreover, patients taking maintenance doses of UST injections every 8 or 12 weeks achieved prolonged disease remission, and the occurrence of undesirable events was comparable to the placebo [42]. UST is equally effective in inducing and sustaining remission for patients with moderate-to-severe UC compared with placebo [43], and future studies are expected to compare the effectiveness of UST with existing biologics in treating UC [44]. Briakinumab, an anti-IL-12p40 mAb, has been terminated from production and follow-up studies due to uncertainty and low confidence in its actual role in the induction and maintenance of clinical remission in CD patients [45].

IL-23 is a pivotal promoter for chronic intestinal inflammation, and targeting IL-23-specific p19 subunits has proven to be effective in phase 2 studies of IBD [46], which may facilitate selective blockade of intestinal lymphatic trafficking without compromising systemic immune defense [47]. There are four mAbs (risankizumab, brazikumab, mirikizumab, and guselkumab) currently applied in clinical trials for either CD or UC [48]. Risankizumab showed curative efficacy and an admissible safety profile in patients suffering moderate-to-severe CD [49], and subcutaneous risankizumab demonstrated maintenance efficacy (including >71% clinical relief and >42% endoscopic remission) in a phase 2 study with good treatment tolerability [50]. Additionally, IL-23/IL-17 axis-related genes were differentially expressed in the transcriptome profile of the ileum and colon of patients after risankizumab treatment [51]. Brazikumab treatment was correlated with clinical improvement in patients with moderate-to-severe CD at 8 and 24 weeks [52]. Mirikizumab exhibited an effective induction of a positive clinical response following 12 weeks of treatment in UC patients (22.6% in the 200 mg group vs. 4.8% in the placebo group) and showed lasting benefits during the entire maintenance period, while it should be noted that the treatment effect of mirikizumab did not seem to proceed in a dose-dependent manner, and further studies are required to determine the optimal dose [53]. Comparatively, approximately 50% of patients initially unresponsive to mirikizumab achieved a clinical response after receiving a prolonged induction period at an increased dose (600 or 1000 mg [54]). Different doses of guselkumab had a good safety profile compared to placebo, achieving better clinical and endoscopic remission (200 mg: 57.4%, 500 mg: 55.6%, and 1200 mg: 45.9% vs. placebo: 16.4%) in a phase 2 clinical trial (GLAXI-1). Additionally, tildrakizumab, a high affinity humanized IgG1κ antibody targeting IL23p19, has proven efficacy in chronic plaque psoriasis and is expected to be implemented into IBD management [55].

Since the effect of IL-23 blockers in the management of chronic inflammatory diseases was first confirmed in psoriasis and subsequently revalidated in other inflammatory diseases, such as IBD, learning from the long-term experience of their applications in other diseases may help to accelerate the optimization of IL-12/23 dosing regimens for IBD patients by gastroenterologists and avoid drug adverse events [56].

#### 2.2.2. IL-6/IL-6R Inhibitors

The engagement of IL-6 in IBD and colorectal cancer pathogenesis has been well explored [57], and some anti-IL-6 mAbs have been tested in RCTs against autoimmune diseases. IL-6 is rapidly produced in response to tissue injury and infection and promotes host immunity by stimulating acute phase responses and immune reactivity, while imbalanced IL-6 synthesis contributes to the pathological effects on chronic inflammation and immune action [58]. IL-6 mediates the proliferation of TH17 cells in concert with TGF-β by affecting signal transducers and activators of transcription (STAT3), which is essential for TH17 cell maintenance and function [59].

PF-04236921, an antibody against IL-6, was investigated among adult CD patients (ANDANTE I and II), with an increased remission rate after a 12-week treatment period (27.4% in PF-04236921 vs. 10.9% in placebo) [60]. PF-04236921 has the potential to induce clinical responses and palliations for refractory CD patients not responding to anti-TNF regimens, but its possible side effects (gastrointestinal abscesses and abdominal pain) warrant attention during subsequent development. Clone MP5-20F3 (an IL-6 mAb) neutralized DSS-induced elevated IL-6 concentrations and inhibited Claudin-2 expression in mice, improving intestinal inflammation and permeability, and its therapeutic effects were further amplified upon coadministration with an anti-TNF-α mAb [61]. An intriguing study on diet and colon cancer demonstrated that a high-calorie diet elevated IL-6 expression in porcine colonic mucosa and changed the expression of IL-6-associated proteins such as PI3KR4, IL-1α, and Map2k1, while a whole food diet inhibited HCD-induced alterations in IL-6-related proteins and modulation of IL-6 signaling was strongly associated with diet-related colitis/cancer [62]. Notably, IL-6 is involved in conditioning critical pharmacokinetic enzymes, such as cytochrome P450s (CYPs), and receiving anti-IL-6 management may restrict the AUCs of drugs with CYP substrates [63], which would impair drug management of many diseases.

Blocking IL-6R also effectively restricts IL-6 complex formation and modulates IL-6-mediated chronic inflammation. Tocilizumab, a humanized IL-6R blocker, is currently approved as a treatment for rheumatoid arthritis and has proven effective in other recalcitrant autoimmune diseases, and its therapeutic applications in IBD are promising [64]. Furthermore, since the chronic proinflammatory activity of IL-6 depends on the gp130 coreceptor for trans-signaling, the decoy protein sgp130Fc (olamkicept) was developed to specifically block IL-6 trans-signaling. Olamkicept resulted in a clinical response by 44% of patients with active IBD in a prospective 2a trial, and its clinical effectiveness was consistent with transcriptional changes in target inhibition [65].

Overall, IL-6 inhibition is a viable treatment option for IBD, but the benefit of completely blocking IL-6 or related receptors is restricted by its strong immunosuppression [65].

#### 2.2.3. IL-1β/IL-1R Antagonists

IL-1β is the main proinflammatory mediator of inflammation. Its secretion is triggered predominantly by macrophages in response to injurious stimuli and it has high expression in IBD patients. The analysis of intestinal biopsy specimens and circulating cytokine profiles from 30 UC patients showed that nearly three-quarters of primary nonresponders to anti-TNF therapy were characterized by hyperexpression of IL-1β in serum and gut tissue [66]. Despite the fact that IL-1β is considered to be as crucial a cytokine as TNF-α in the onset of IBD, there are currently few specific antibodies against IL-1β, excluding canakinumab [67]. The absence of binding to murine-derived IL-1β hindered preclinical studies of canakinumab [68].

A novel chimeric anti-IL-1β-specific mAb, 7F IgG, with high affinity for various mammalian IL-1βs, blocked IL-1β to regulate proinflammatory cytokines and showed anticolitis activity in a TNBS-induced murine model [68]. Antagonism of the IL-1 receptor (IL-1R) is a therapeutic alternative to reduce the IL-1β stimulatory response. Endogenous IL-1R antagonists have been discovered to be important in host defense against overwhelming endotoxin-induced injury, and they are generated in experimental animal models of multiple diseases and in human autoimmune inflammatory diseases and are essential natural anti-inflammatory proteins in colitis [69]. Anakinra (an IL-1R antagonist) treatment ameliorated dinitrobenzene sulfonate (DNBS)-induced colitis in terms of macroscopic and histological changes, inflammatory cell infiltration, and oxidative stress [70]. Two IL10R-deficient patients with severe refractory IBD had significant clinical, endoscopic, and histological responses following 4–7 weeks of anakinra therapy [71]. A phase 2, multicenter RCT (IASO) of short-term anakinra treatment in patients with acute severe ulcerative colitis (ASUC) has been approved by the Cambridge Central Ethics Committee, a clinical trial has been authorized [72], and the prognostic outcome of using anakinra to intervene in IL-1 signaling in patients with ASUC is of ongoing interest. A proportion of de-N-acetylated oligomers (13 < dp < 20) were reported to rescue inflammatory impairments via an IL-1Ra-dependent pathway, and water-soluble galactosaminogalactan oligosaccharides might be therapeutically applicable as new anti-inflammatory glycosides in IL-1-mediated hyperinflammatory diseases [73].

Indeed, the release of IL-1β is driven by the activated NOD-like receptor family pyrin domain containing 3 (NLRP3) inflammasome, which is harmful in colitis [74]. In a comparative preclinical study of NLRP3 inflammasome signaling inhibitors, it was found that administration of INF39 (an NLRP3 inflammasome irreversible direct blocker), Ac-YVAD-cmk (a caspase-1 inhibitor), and anakinra to rats with DNBS-induced colitis resulted in colitis relief and that INF39 was more effective in reducing organizational increments of myeloperoxidase, TNF, and IL-1β, and attenuating intestinal inflammation [75]. OLT1177, a selective inhibitor of NLRP3 inflammasomes, was administered during the induction period of DSS-induced mice to alleviate the disease phenotype, and OLT1177 was effective in preventing the onset of DSS colitis. However, the administration of OLT1177 during the recovery phase of colitis did not promote the recovery course or tissue remission [75]. Curcumin potently inhibited NLRP3 inflammasome activation to alleviate colitis in a DSS-induced murine model, and the therapeutic outcome was partially limited following the use of MCC950 (a specific NLRP3 blocker) [76].

#### 2.2.4. Other Interleukin Neutralizers with Potential Efficacy

Experimental animal studies and human genetic studies are gradually revealing the importance of the interleukin superfamily in regulating inflammation and tissue damage in the intestinal mucosa [77]. More members of the interleukin family are being noticed in colitis, and targeted therapies are being explored.

IL-18 is a proinflammatory interleukin matured by NLRP3 inflammasome activation, and IL-18 equilibrium in the epithelium has been implicated in barrier function in colitis, where IL-18 coordinates goblet cell maturation transcriptional programs [78]. Mokry conducted a Mendelian randomization study of 12,882 cases and 21,770 controls to examine the effect of elevated IL-18 on IBD susceptibility, and the results revealed that each genetically predicted SNP change (rs385076, rs17229943, and rs71478720) in IL-18 was associated with elevated susceptibility to IBD [79]. GSK1070806, a neo-IgG1 mAb that neutralizes IL-18, has been tested in a pilot trial in type 2 diabetes and kidney transplantation [80], and a corresponding RCT based on genomic detection of IL-18 in IBD is warranted.

Overproduction of IL-13 in active UC causes disturbances in the intestinal epithelium and may promote fibrosis. Anrukinzumab is a humanized IgG1 antibody conjugated to IL-13 that impedes the binding of IL-13 and IL-4Rα without affecting the attachment of IL-13 to IL-13α1/α2. After receiving anrukinzumab intravenously, the total serum level of IL-13 increased (free IL-13 binds to anrukinzumab) in patients with active UC, while changes in fecal calprotectin at 14 weeks posttreatment were not significantly different compared to placebo, and the treatment effect did not have a statistically significant difference [81]. Tralokinumab, an IL-13 neutralizing antibody, failed to induce a clinical response when used as an add-on therapy randomized to ambulatory adult UC patients, whereas tralokinumab had higher rates of clinical remission (18% vs. 6%) and mucosal healing (32% vs. 20%) than placebo, suggesting that it may benefit some UC patients and be well tolerated [82]. Likewise, although IL-17A is an active cytokine in gut inflammation, the clinical outcome of anti-IL-17A therapy (secukinumab and ixekizumab, anti-IL-17A mAbs) in CD patients is not favorable [83], and a study based on IL-17a-/- mice found that blockade of IL-17A may cause IL-6 upregulation and RORγt+ ILC recruitment during chronic colitis, leading to elevated IL-22 and even worsened disease [84].

Apart from the interleukins mentioned above, the role played by other interleukins, such as IL-5 [85], IL-21 [86], and IL-33 [87], in IBD development is being uncovered as the disease mechanism comes to be better understood, and subsequent drug development for them is anticipated. The network of interleukin family actions is intertwined, and a complete blockade of a singular interleukin may lead to systemic involvement. This may explain the involvement of some interleukin family members in mediating IBD and the lack of efficacy in targeted antagonistic therapy [88]. Long-term follow-up after anti-interleukin control and regular monitoring of relevant therapeutic indicators are necessary to avoid potential adverse drug effects.

### 2.3. Anti-Interferon Agents

Interferons (IFNs) are an essential family of immunostimulatory cytokines, classified broadly into three subtypes: type I (IFN-α, β, ε, κ and ω), type II (IFN-γ), and the recently defined type III (IFN-λ) [89], among which INF-γ has been extensively studied in IBD. IFN-γ upregulation is common in IBD, and IFN-γ is vital for mucosal barrier defense function and stimulates immunomodulatory signaling. Increased populations of INF-γ-secreting cells in the lamina propria were identified as correlating with a higher disease activity index in CD [90], and a population-based screening cohort study of plasma samples identified IFN-γ as a potential key regulator of UC [91]. The pathogenic effect of IFN-γ on IBD may not only involve its immunomodulatory effects but also depend on its participation in the disruption of the vascular barrier [92]. The profile of serum type Ⅰ and type Ⅱ IFN in IBD patients receiving anti-TNF therapy correlated significantly with the treatment response in IBD patients and could be regarded as a clinical biomarker [93]. Excess type I interferon signaling in the intestinal epithelium is involved in intestinal panniculocyte dysfunction, and the Western diet is one of the environmental triggers leading to this process and ultimately to the pathogenesis of CD [94]. Recently, IFN-λ has also been proven to be involved in IBD, disrupting colonic epithelial healing and inducing small intestinal epithelial death, although associated studies are at a preliminary stage with complex and “paradoxical” effects of IFN-λ [95].

Fontolizumab, an anti-IFN-γ antibody, demonstrated efficacy in individuals with refractory CD, in which patients received two doses of fontolizumab intravenously followed by increased treatment response rates and remission on day 56, whereas clinical remission on day 28 after a single dose of fontolizumab was not significant [96]. Unfortunately, there have been no subsequent reports of fontolizumab for the treatment of colitis in recent years. Although quite a few biologics with activity against type Ⅰ and Ⅱ interferons are currently available, such as sifalumumab, rontalizumab, and anifrolumab (some of which have been approved for treating other immune diseases) [97], very few studies have been performed for the treatment of colitis. This is inextricably bound up with the double-sided action of IFN for colitis, and IFN profiling and other biomarkers in the serum of clinical patients could help to find suitable recipients for such therapies. Alternatively, small molecules of plant origin could be a potential breakthrough, offering a more “moderate” inhibitory activity against interferon than monoclonal antibodies. For example, feeding berberine inhibits IFN-γ and IL-17A in SCID mice administered with CD4+CD45RB high T cells, reduces lamina propria lymphocyte infiltration, and improves colitis [98].

### 2.4. Janus Kinase Inhibitors

Activation of Janus kinases (JAKs) is required for cytokine signaling and mediates the binding of cytokines and receptors. Consequently, blocking JAKs may be an efficient way to simultaneously modulate the activity of multiple proinflammatory cytokines to improve colitis [99].

Tofacitinib, a small-molecule JAK suppressor, is currently a promising medication for IBD and is approved for UC management [99]. Tofacitinib was superior to placebo for both inducing and maintaining treatment in three phase 3 clinical trials (OCTAVE) for UC. Clinical remission occurred in 18.5% of patients treated with 10 mg tofacitinib for 8 weeks and 8.2% of the placebo group, and similar improvements in treatment outcomes were reached in the other parallel trials. At 52 weeks, the maintenance effect of tofacitinib was apparent, as the remission rates in the 5 mg tofacitinib, 10 mg tofacitinib, and placebo groups were 34.3%, 40.6%, and 11.1%, respectively [100]. However, the risk of infection and cardiovascular events emerging from this study are of ongoing concern for follow-up. Tofacitinib is a rescue option for TNF inhibitor treatment failures, and an additional 10 mg of tofacitinib treatment usually leads to positive clinical responses for most patients [101]. In addition, recent studies have provided evidence of endoscopic and tissue remissions with tofacitinib for patients with refractory UC [102]. Nevertheless, the therapeutic benefit of tofacitinib on CD was not as pronounced as that on UC, and the results of several CD-related studies [103,104] demonstrated that the primary efficacy endpoint of tofacitinib was fundamentally similar to that of a placebo.

## 3. Targeting Leukocyte Trafficking

Leukocyte trafficking in the gut is strictly coordinated and engaged in maintaining intestinal immune equilibrium, mediating immune responses, and modulating inflammation, while in cases of colitis, large numbers of leukocytes migrate through the lymphatic and blood circulation into the gut and are activated, which is considered a vital step for a flare-up of IBD [105]. With the success of the anti-α4β7 antibody vedolizumab, the concept of addressing leukocyte homing for IBD management has been highlighted, which has become an exciting pillar of IBD therapy [106]. Leukocyte recruitment in inflammation is a complicated and sophisticated process requiring the binding of chemoattractants and receptors, the recognition of integrins, and the fastening of adhesion molecules. Thus, this section presents an overview of IBD agents targeting leukocyte trafficking (Figure 3), mainly including integrins, cell adhesion molecules (CAMs), chemokines, selectins, and sphingosine 1-phosphate (S1P) signaling.

### 3.1. Integrins

The integrin family is of particular importance in lymphatic homing, and various integrins expressed on endothelial cells are fundamental to this process. Integrins expressed on specific lymphocyte subpopulations are implicated in the onset of IBD [107]. Briefly, integrins are glycoprotein receptors present on the cell surface and are comprised of heterodimeric α and β subunits, to which CAMs and extracellular matrix bind [108]. Morphological alterations of integrins sparked by external stimuli (e.g., cytokines) boost their avidity for their corresponding ligands and guide the migration of lymphocytes. As an example, the integrins α4β1 and α4β7 enhance the translation of proinflammatory T lymphocytes into the lamina propria. Subsequently, cytotoxic T lymphocytes are preserved by the interaction between αEβ7 and E-cadherin once they have entered the lamina propria [109].

#### 3.1.1. Anti-α4 Antibody

α4 was the first integrin explored as a potential target for the management of UC and CD, and as a common subunit of α4β1 and α4β7, HP1/2 (an anti-α4 mAb) significantly alleviated acute colitis in a cotton-top tamarin spontaneous colitis model [110]. Consequently, blocking leukocyte–vascular adhesion and other integrin-mediated IBD management options have been acknowledged and developed rapidly over the past three decades.

Natalizumab (a humanized α4 antibody) induced and maintained remission in refractory CD patients in a phase 3 trial, but its increased risk of progressive multifocal leukoencephalopathy was unacceptable; therefore, it has basically been withdrawn from clinical use [111]. Nonintestinal selective anti-α4 mAb impairs leukocyte recycling of the central nervous system, and the oral small-molecule α4 inhibitor AJM300 is currently the most promising agent. After receiving oral AJM300, 102 patients with moderately active UC presented with significantly higher clinical response rates (62.7% vs. 25.5%), clinical remission rates (23.5% vs. 3.9%), and mucosal healing rates (58.8% vs. 29.4%) at week 8 compared to the placebo and were free of serious adverse events [112]. The effectiveness and safety of AJM300 were recently reaffirmed in a double-blind phase 3 study, in which 45% of UC patients who did not respond to mesalazine intervention achieved clinical remission after AJM300 intervention (compared to 21% in the placebo), while the frequency of adverse incidents was comparable to placebo [113].

#### 3.1.2. Anti-α4β7 Antibody

α4β7 integrin conjugates with its ligand MAdCAM-1 to accomplish intercellular anchoring, and since MAdCAM-1 is expressed mostly within the intestine, anti-α4β7 antagonism is deemed to be gut-selective, circumventing the potential risk of systemic inhibition by anti-α4 therapies.

Vedolizumab, a selective anti-α4β7 integrin mAb, achieved improved and durable clinical responses in refractory UC patients in a phase 3 double-dummy trial with both subcutaneous and intravenous administration [114]. In a phase 3b head-to-head clinical trial, vedolizumab was preferable to adalimumab in terms of attaining clinical remission and endoscopic improvement [115]. Additionally, vedolizumab had the best safety profile according to a network meta-analysis of small-molecule drugs and biologics for UC [116]. The efficacy of vedolizumab in CD patients is also remarkable, given that in a prospective trial, approximately one-third of active CD patients reached endoscopic remission at week 52 after treatment with vedolizumab, and two-thirds of patients attained histological remission at week 26 [116]. In a phase 3 RCT (VISIBLE 2), a new subcutaneous formulation of vedolizumab was reported to be active and safe in the maintenance treatment of adults with active CD [117]. Abrilumab, a fully human anti-α4β7 mAb, was investigated in a phase 2 trial in Japanese refractory UC patients (receiving conventional therapy) and exhibited superior clinical remission outcomes relative to placebo at week 8 [118]. PTG-100, an oral α4β7 antagonist peptide, has shown potential for IBD treatment in both preclinical models and phase 2a trials in UC patients, effectively inhibiting memory T-cell trafficking with a good safety profile [119].

#### 3.1.3. Anti-β7 Antibody

Selective antagonism of the cosubunit β7 in α4β7 and αEβ7 integrins is another anti-integrin strategy being investigated. Etrolizumab, an anti-β7 integrin mAb, markedly induced clinical remission among patients suffering active UC in an early phase 2 study, while in a recent phase 3 trial (HICKORY), etrolizumab significantly increased the remission proportion at week 14 in refractory UC patients who had already received anti-TNF therapy [120]. A double-dummy phase 3 RCT (GARDENIA) involving 397 active UC patients demonstrated that etrolizumab had similar activity as infliximab in terms of improving clinical remission from a clinical perspective [121]. The efficacy of etrolizumab in CD patients was assessed in an RCT (BERGAMOT) and the open-label extension trial JUNIPER. A higher rate of clinical endoscopic remission was achieved with etrolizumab compared to placebo, suggesting that blocking α4β7 and αEβ7 may be beneficial in the refractory CD population [122].

### 3.2. Inhibitors of Cell Adhesion Molecules

Cell adhesion molecules are overexpressed in inflamed mucosa, and their binding to integrins is responsible for leukocyte recruitment, making targeting of CAMs attractive [123].

MAdCAM, a member of the immunoglobulin superfamily, influences the adhesion of circulating leukocytes to inflamed areas and regulates lymphocyte infiltration [124]. Immunohistochemical analysis of colonic biopsies from UC patients showed that MAdCAM-1 expression persisted after venous inflammation had subsided and then it increased in subsequent episodes, possibly as a trigger for disease recurrence and as a chronic therapeutic target [125]. The efficacy and safety of ontamalimab for UC, a fully human mAb against MAdCAM-1, was preliminarily confirmed, and ontamalimab was well tolerated by patients in the long term (144 weeks) with acceptable adverse events [126].

Intercellular adhesion molecule 1 (ICAM-1), as a hyperglycosylated transmembrane protein, is expressed on nonhematopoietic cells and is rapidly upregulated upon exposure to inflammatory stimuli [127], whereas the expression pattern of ICAM-1 in IBD implies a relevant pathological role [128]. Alicaforsen, an anti-ICAM-1 antisense oligonucleotide, shows only mild efficacy in IBD by inhibiting the translation of ICAM-1 mRNA [128]. Although alicaforsen did not appear therapeutically effective in phase Ⅱ trials for parenteral application, its efficacy in distal UC as a topical enema was improved [129]. In addition, increased expression of multiple CAMs (PECAM-1, ICAM-3, and VCAM-1) in colon biopsies from quiescent IBD patients is suspected to be associated with subsequent disease onset.

Collectively, although CAMs have been progressively confirmed to participate in IBD pathogenesis, current treatment options for CAMs are limited and do not achieve the desired results.

### 3.3. Regulation of Chemokines

Chemokines, namely, chemotactic cytokines, regulate the movement and localization of peripheral immune cells in tissues, directing activated cells to drain lymph nodes to initiate and imprint adaptive immune responses [130]. A range of chemokines is distinctly increased in IBD patients compared with normal healthy donors [131], and chemokines and related receptors are potential targets for IBD treatment.

CCL20 is a member of the CC chemokine superfamily, constitutively expressed by intestinal epithelial cells, whose production is markedly increased in CD patients [132]. CCL20 inhibits TGF-β1-induced Treg (iTreg) differentiation and predisposes cells toward the pathogenic TH17 lineage in a CCR6-dependent manner [133]. The subtle equilibrium between TH17 and Treg cells maintained by the CCL20/CCR6 axis is disrupted under inflammatory states, violating mucosal tolerance. Genome-wide association studies in IBD patients have indicated a robust correlation between CCR6 expression and disease severity, and blockade of CCR6 or CCL20 with antagonists provides relief by preventing the trafficking of CCR6-expressing lymphocytes at sites of inflammation [134]. *Taraxacum officinale* extract significantly reduced the expression of CCL20, CCR6, and CXCL1/5 in DSS-induced mice and alleviated colitis by modulating fatty acid metabolism and microorganisms [135]. After two weeks of treatment with mongersen (an anti-Smad7 oligonucleotide), a proportion of active CD patients responded with reduced serum levels of CCL20, which may be a potential treatment modality [132]. PRN694 significantly hindered the development of T-cell-driven adoptive transfer model colitis and T-cell infiltration in tissues, which may be associated with impaired CXCL11 and CCL20 migration [136]. BI119, a novel oral RORγt antagonist, downregulated the expression of CXCL1, CXCL8, and CCL20 in intestinal crypt cultures in vitro, and BI119 exhibited the same regulatory effect in a T-cell transfer animal model [137].

CXCL10 promotes the recruitment of neutrophils, T-cell activation, and secretion of downstream cytokines. Eldelumab, a fully human anti-CXCL10 mAb, showed a higher clinical remission and response rate in anti-TNF-primed UC patients, despite not meeting the primary induction treatment endpoint in a placebo-controlled phase 2b study [138]. Induction therapy with eldelumab in active CD exhibited clinical and endoscopic efficacy trends, without an observed exposure–remission relationship for eldelumab [139].

Serum CCL11 is significantly increased in IBD patients, considered to be associated with increased colonic eosinophils, and Ccl11-/- mice were found to be tolerant to DSS-induced colitis and to reduce the colonic tumor load induced by azomethine (AOM)-DSS, suggesting that CCL11 antibodies might be valuable for IBD treatment and cancer prevention [140]. As research progresses, a growing list of chemokines are being identified as relevant to IBD progression or as disease-monitoring indicators, but the development of anti-chemotactic agents still needs further optimization. For instance, the poor pharmacokinetics of vercirnon (a prototype CCR9 antagonist) might cause invalid clinical trial results, and oral CCR antagonists based on the 1,3-dioxoindoline backbone could offer a possible direction for optimization [141].

### 3.4. Selectins

Selectins are an adhesion molecule family comprising L-, E-, and P-selectins that primarily bridge the interactions between leukocytes and the endothelium [142]. L-selectin is widely represented by leukocytes, whereas E-selectin and P-selectin are mainly expressed on the surface of endothelial cells exposed to inflammatory stimuli. Selectin deficiency or dysfunction contributes to inflammatory diseases, and targeting selectins in IBD has been proven to be feasible. P-selectin exerts its physiological function by binding to P-selectin glycoprotein ligands (PSGL-1). The serum concentration of soluble PSGL-1 in UC patients is significantly increased compared with that in controls, and the anti-PSGL-1 antibody resulted in a short-term blockade of neutrophil trafficking and reduced UC activity [143]. L-selectin is a promising independent predictor of anti-TNF management based on a 5-year clinical follow-up of IBD, aiding patients in personalizing treatment and reducing the risk of adverse events [144].

### 3.5. Modulation of Sphingosine 1-Phosphate Signaling

Sphingosine-1-phosphate (S1P) signaling is a promising target for immune cell homing, which regulates lymphocyte efflux into recirculation and the accumulation of lymphocytes in inflamed intestinal segments [145]. Although anti-S1P mAb (Sphingomab) has shown encouraging therapeutic results in maintaining inflammatory endothelial barrier function, no study has been conducted for IBD.

Antagonizing S1P receptors is the prevailing research direction for interrupting S1P signaling. Fingolimod, the first FDA-approved S1P receptor immunomodulator for multiple sclerosis treatment, has shown symptomatic relief in multiple experimental murine colitis models [145]. However, fingolimod has been proven to cause lymphopenia and has potential cardiac and vascular side effects. Currently, of greater interest is a novel class of oral small-molecule S1P receptor modulators.

Ozanimod, as a selective S1P receptor subtype 1 and 5 modulator, offers great induction and the maintenance of therapeutic efficacy in IBD treatment. In a multicenter phase 3 study of patients with moderate-to-severe active UC, the proportion of patients achieving clinical remission with ozanimod treatment in the induction phase was 16% compared to 6% for placebo, and 37% of patients achieved clinical remission with ozanimod treatment in the maintenance phase compared to 18.5% for placebo [146]. Meanwhile, during the 52-week trial, the incidence of infection with ozanimod was similar to placebo, with serious infections occurring in extremely few patients. A 4-year follow-up of the phase 2 TOUCHSTONE trial showed that the tolerability and long-term benefit rates of ozanimod administration were substantial, with 93.3% and 82.7% clinical response and remission rates at week 200, respectively, based on Mayo measurements, and no new safety risks were observed during the follow-up period [147]. Ozanimod has also contributed to treating refractory CD. Endoscopic improvement was observed in 23.2% of patients, clinical remission in 39.1%, and treatment response in 56.5% of patients at 12 weeks after receiving ozanimod. In addition, relevant phase 3 placebo-controlled trials are under investigation [148].

Etrasimod, a selective S1P1 receptor modulator, is being evaluated for immune-mediated inflammatory diseases. Patients with refractory UC met the primary and all secondary endpoints of the trial when given a daily oral dose of 2 mg of etrasimod at week 12. Endoscopic improvement was reported in 41.8% of patients receiving etrasimod versus 17.8% of patients receiving placebo [149]. A long-term efficacy study on etrasimod for UC treatment (OASIS) described the favorable therapeutic effect and safety of etrasimod in UC patients, approximately half of whom had a clinical response and endoscopic remission over 12 weeks of continuous treatment [150]. Although the vast majority of adverse events occurring during 52 weeks of etrasimod treatment were acceptably mild/moderate, such as anemia and exacerbation of UC, appropriate caution could not be exercised regarding drug safety monitoring.

Amiselimod, an oral S1P1 receptor modulator, may have a better safety profile than other S1P receptor modulators, given its lower EC50 value. After 3 days of continuous amiselimod administration, the expression of the S1P1 receptor on mesenteric lymph node T cells in mice was almost completely eliminated, and amiseimod inhibited disease progression and T-cell infiltration in the lamina propria in the T-cell adoptive transfer colitis model [151]. However, amiselimod did not show a marked advantage over the placebo group in terms of induction of a clinical response in a recent phase 2a trial on refractory CD patients [152].

## 4. Discussion

Cytokines secreted by activated lymphocytes lead to damage to the mucosal barrier and increased inflammation, participating in the pathogenesis of IBD [4]. Targeting proinflammatory cytokines and lymphocyte trafficking in IBD is a comparatively flawless therapeutic strategy [5]. Since more biological and small-molecule medications are available, both patients and clinicians are challenged to choose the “right” drug. Therefore, in this review, the recent targets and agents aimed at cytokines and immune cell homing in IBD are summarized to assist in understanding the current status of drug development.

New treatment strategies offer extended options for IBD treatment. Steroid-free remission is one of the main goals of IBD treatment due to the adverse effects of corticosteroids, and prolonged usage of corticosteroids is a sign of unsatisfactory care [153]. The annual use of corticosteroids has indeed declined yearly with the emergence of new biological therapies [154]. Over the past few decades, anti-TNF agents and vedolizumab have been adopted for clinical IBD treatment, and facilitate the reduction of corticosteroid use and improve the response and remission rates [155]. However, no guidance is currently available on the most suitable second-line treatment after the failure of anti-TNF therapy [156]. The novel therapeutic agents listed herein combined with the monitoring of disease-related biomarkers during treatment will enlarge our existing therapeutic armamentarium to achieve deep remission of IBD [157]. Cytokines and cell trafficking blockers have yielded promising results facilitating individualized treatment of IBD. Head-to-head comparative trials are now required to supply proof for choosing appropriate medications. Moreover, a more proactive attitude is needed toward emerging agents. The preference of patients and providers for “relatively safe and effective” standard therapies has led to a dilemma in the usage of new medications. The proportion of patients receiving first-line combination therapy (any biologic therapy with any immunomodulator) was less than 1% [158]. With the drugs available, there are more opportunities to greatly improve the quality of life of patients [159].

Overall, selective inhibition of proinflammatory cytokines provides expanded options for achieving clinical remission in IBD [160], while blockade of lymphocyte homing may be a more effective option for modulating proinflammatory mediators. Treatment follow-up and drug safety evaluation will facilitate the subsequent development and clinical application of these agents.

## Figures and Tables

**Figure 1 pharmaceuticals-15-01080-f001:**
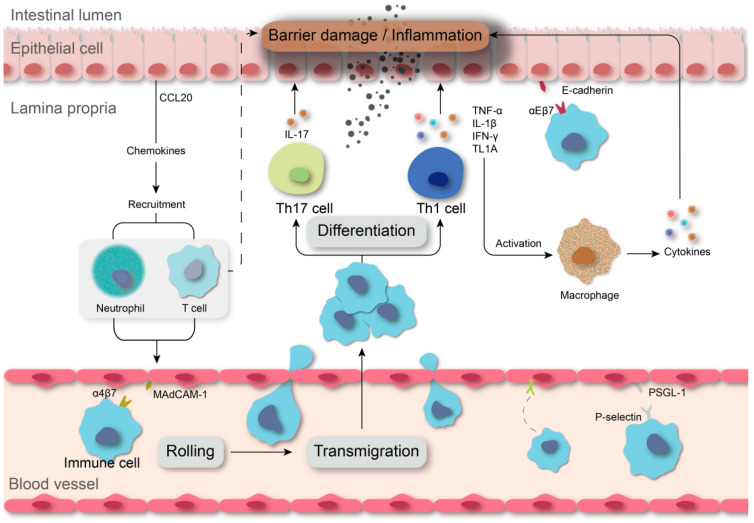
Immune cell homing and cytokine storming drive IBD progression. Lymphocytes roll, adhere, and transmigrate at the site of the lesion through the combined action of integrins, selections, and chemokines, with residence in the lamina propria. Excessive immune responses and external stimulus synergistically lead to a proinflammatory cytokine storm hindering disease recovery.

**Figure 2 pharmaceuticals-15-01080-f002:**
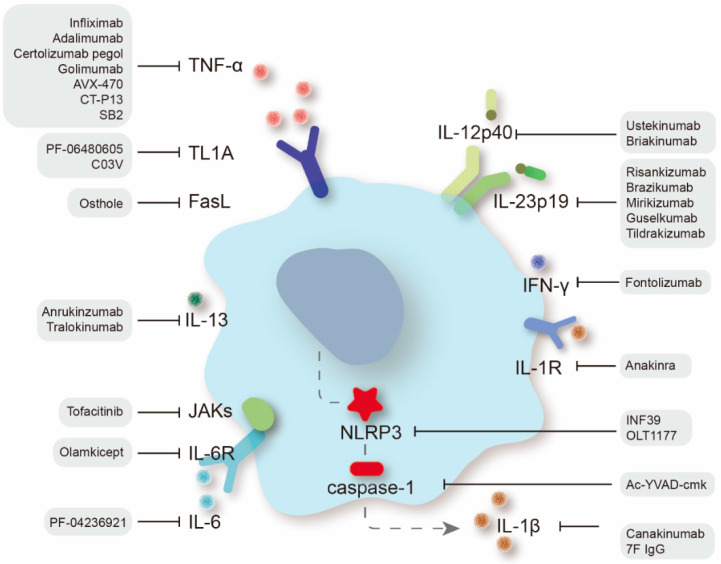
Therapeutic targets of proinflammatory cytokines in IBD treatment and specific therapeutic agents.

**Figure 3 pharmaceuticals-15-01080-f003:**
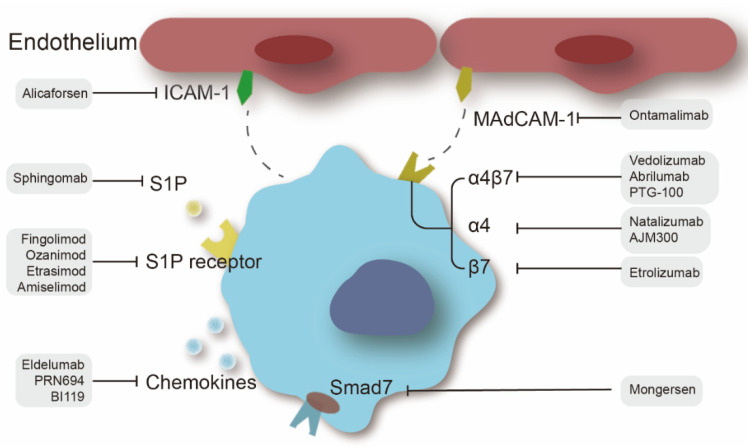
Therapeutic targets of leukocyte trafficking in IBD treatment and specific therapeutic agents.

## Data Availability

Not applicable.

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
