# Peer review of "Tackling Inflammatory Bowel Diseases: Targeting Proinflammatory Cytokines and Lymphocyte Homing"

_pharmaceuticals, 2022, doi:10.3390/ph15091080_

Round 1
Reviewer 1 Report
This is a review article that describes the potential role of cytokines and immune cell infiltration in IBD, with a possible implication for drug development. The manuscript is well-written with minimal mistakes in the text. The figures are self-explanatory. I have only some minor comments:
- Figure 1. Inflmmation it should be Inflammation;
- Section 2.1.1. Anti-TNF-α agents. The results of the SICURA project should be included (Antioxidants (Basel). 2022 Jan 8;11(1):138. doi: 10.3390/antiox11010138.)
Author Response
Thank you very much for your valuable comments on our manuscripts.
-Our thanks for pointing out the minor error we missed, and we have corrected the word "inflammation" in Figure 1.
-Antioxidant-rich nutritional supplementation is indeed a powerful facilitator to anti-TNF therapy, hence SICURA project is included in lines 158-160. The details in the article are as follows.
"In addition, complementing anti-TNF therapy with nutritional biocompounds, such as antioxidant-enriched purple corn supplement [38], has exhibited beneficial effects on the induction and maintenance of IBD remission."

Reviewer 2 Report
The review, Tackling inflammatory bowel diseases: targeting proinflammatory cytokines and lymphocyte homing, lives up to its promise to assemble information on major targets for potential IBD therapy and treatment. I think the title of the article should be reworded, to attract a greater audience.
Reviewer 3 Report
This review article is very interesting and worth. I really enjoined this article while reading. There is no portion that require revision.
Author Response
We sincerely thank you for your appreciation of this article and the time you devoted to reviewing this manuscript.